# Structural basis of synthetic agonist activation of the nuclear receptor REV-ERB

Meghan H. Murray[1,2], Aurore Cecile Valfort[3], Thomas Koelblen[3], Céline Ronin[4], Fabrice Ciesielski[4], Arindam Chatterjee ⓘ [1], Giri Babu Veerakanellore[2,5], Bahaa Elgendy ⓘ [2,5], John K. Walker[1], Lamees Hegazy[2,5] ✉ & Thomas P. Burris ⓘ [3] ✉

The nuclear receptor REV-ERB plays an important role in a range of physiological processes. REV-ERB behaves as a ligand-dependent transcriptional repressor and heme has been identified as a physiological agonist. Our current understanding of how ligands bind to and regulate transcriptional repression by REV-ERB is based on the structure of heme bound to REV-ERB. However, porphyrin (heme) analogues have been avoided as a source of synthetic agonists due to the wide range of heme binding proteins and potential pleotropic effects. How non-porphyrin synthetic agonists bind to and regulate REV-ERB has not yet been defined. Here, we characterize a high affinity synthetic REV-ERB agonist, STL1267, and describe its mechanism of binding to REV-ERB as well as the method by which it recruits transcriptional corepressor both of which are unique and distinct from that of heme-bound REV-ERB.

The nuclear receptor REV-ERB is a component of the molecular clock that regulates the rhythmic expression of inflammatory[1–4], metabolic[5,6], and cellular proliferative genes[7–10]. Genes directly regulated by either REV-ERB, REV-ERBα [NR1D1] or REV-ERBβ [NR1D2], are typically transcriptionally silenced by the receptors. This is due to REV-ERB functioning as a ligand-dependent transcriptional repressor with heme as the endogenous ligand[11,12]. Heme is present in all cells at a concentration that allows REV-ERB to maintain a basal level of repression of target genes providing the appearance of constitutive transcriptional repressor that was initially prescribed to it prior to the discovery of heme as a ligand. REV-ERBs are particularly efficacious transcriptional repressors, and they lack the carboxy-terminal activation function 2 domain required for coactivator protein recruitment. Thus, REV-ERBs display a particularly strong level of recruitment of the corepressor NCoR. REV-ERBs have been demonstrated to play a role in a range of disease states suggesting that agonists may be useful in treating diseases associated with inflammation, metabolic dysfunction, and cancer[13].

The REV-ERB LBD displays the canonical NR structure composed of a three-layered α-helical sandwich, and heme binds in the prototypical ligand binding pocket of the LBD. Quite interestingly, the iron atom within the protoporphyrin ring is coordinated by histidine and cysteine residues that are conserved in both REV-ERBα and REV-ERBβ[14,15]. These two residues had previously been predicted to play such a role in heme binding to REV-ERB and mutation of either of these residues leads to loss of heme binding and loss of transcriptional repressor activity[11,12]. Most recent structural studies have demonstrated that heme binding drives cooperative binding of NCoR interaction domain (ID) peptides to the REV-ERB LBD providing the molecular mechanism for ligand-dependent transcriptional repression by REV-ERB[15].

Heme is an endogenous ligand for REV-ERB[11,12], but since heme binds to many types of proteins, specificity concerns prevent its structure from being used as starting point for synthetic ligand development. The first synthetic ligand to REV-ERB was discovered through a fluorescence resonance energy transfer (FRET) assay showing that a compound increased binding of nuclear co-repressor NR interaction domain 1 (NCoR ID1) to REV-ERBα ligand binding domain (LBD) in a concentration-dependent manner[16]. This initial

[1]Department of Pharmacology and Physiology, Saint Louis University School of Medicine, St. Louis, MO 63104, USA. [2]Center for Clinical Pharmacology, Washington University School of Medicine, University of Health Sciences & Pharmacy, St. Louis, MO 63110, USA. [3]University of Florida Genetics Institute, Gainesville, FL 32610, USA. [4]NovAliX SAS, Strasbourg, France. [5]Department of Pharmaceutical and Administrative Sciences, University of Health Sciences & Pharmacy, St. Louis, MO 63110, USA. ✉e-mail: Lamees.Hegazy@uhsp.edu; burris.thomas@ufl.edu

synthetic REV-ERB agonist, GSK4112, has low potency and efficacy as well as very poor systemic exposure in vivo providing only limited utility as a chemical tool to probe REV-ERB function. Our lab as well as others attempted to improve the pharmacological properties of GSK4112 in order to provide REV-ERB chemical tools with more favorable properties. One of the identified compounds was SR9009[6]. This compound displayed improved potency for REV-ERBα and REV-ERBβ over GSK4112 and specificity for REV-ERB over the other 46 human nuclear receptors. SR9009 is three to four times more potent and three times more efficacious than GSK4112 in a cell-based cotransfection assay, and it also has increased systemic exposure, including the ability to cross the blood-brain barrier[13]. SR9009 has now become the most widely published tool compound for modulating REV-ERB both in cell culture and in vivo.

Although SR9009 and structurally related compounds are useful chemical tools for research, this scaffold is less than ideal due to concerns about relatively poor potency, lack of bioavailablity, potential toxicity, and arguably, specificity. In addition, SR9009 and almost all its analogs contain a nitrothiophene group, which is considered a toxicity liability[13]. A recent study has questioned the specificity of SR9009, calling into question the results from numerous studies using this tool compound to modulate the REV-ERB nuclear receptor[17]. Thus, a scaffold for REV-ERB agonists, one with greater potency, less toxicity, and unquestionable specificity is needed.

Here, we describe the activity of a synthetic REV-ERB agonist with a distinct chemical scaffold relative to SR9009 with improved potency and reduced cell toxicity. Furthermore, we define how the synthetic agonist binds within the ligand binding domain (LBD) of REV-ERB by analyzing the co-crystal structure of the agonist, STL1267, within the LBD of REV-ERBα.

## Results

### STL1267 binds directly to the ligand binding domain (LBD) of REV-ERB

Our previous work on the structure activity relationship of GSK4112/SR9009 chemical scaffold (Fig. 1a) as well as that of others[18–20], led us to seek other scaffolds that would provide improved properties. Utilization of heme as a REV-ERB ligand in various biochemical assays is limited due to its chemical-physical properties[15] and SR9009 analogues also did not provide optimal performance for the development of radioligand binding assays or for the determination of cocrystal structures with REV-ERB to assess the mechanism of synthetic agonist binding to the receptor. We noted a recent patent that described a series of REV-ERB agonists (6-substituted[1,2,4]triazolo[4,3-b]pyridazines) in assays that detected the ability of small molecules to enhance the interaction of REV-ERBα with NCoR either in biochemical or cell-based assays[21]. One of these compounds (we termed as STL1267; Fig. 1b; Supplementary Fig. 1, synthesis scheme can be found in Supplementary Methods), displayed potency of ~0.1–0.3 μM in these assays and we decided to characterize it more fully as shown below. A FRET assay that detects agonist-dependent recruitment of NCoR ID1 CoRNR box to the REV-ERBα-LBD was utilized to compare the activity of STL1267 to SR9009 (Fig. 1c). We observed dose-dependent recruitment of NCoR peptide with increasing amounts of either STL1267 or SR9009. The efficacy of NCoR ID1 recruitment in the presence of STL1267 was roughly double that of SR9009, and the potency of STL1267 ($EC_{50} = 0.13\,\mu M$) was more than ten times greater than SR9009 ($EC_{50} = 1.9\,\mu M$) (Fig. 1c). The high potency of STL1267 led us to consider the development of a radioligand binding assay that would detect direct binding of ligands to the LBD of REV-ERBα and REV-ERBβ.

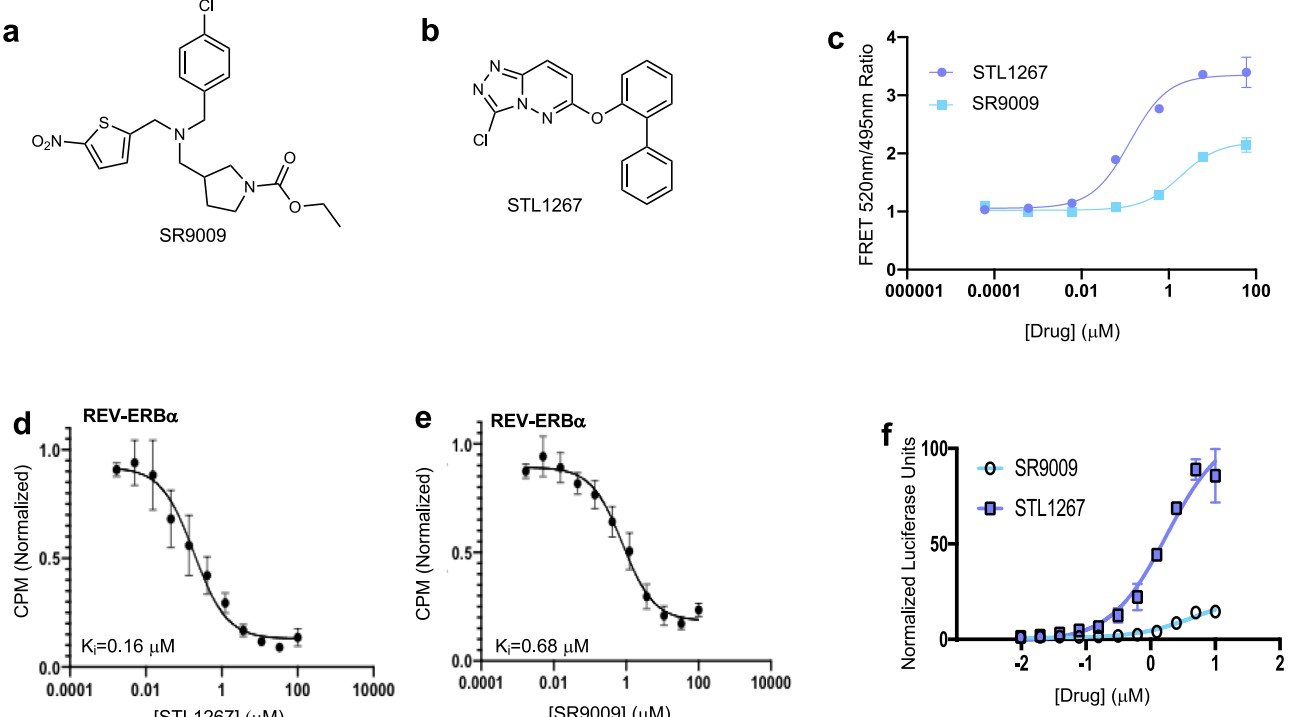

**Fig. 1 | STL1267 binds to REV-ERB, resulting the recruitment of NCoR and regulation of downstream target genes.** Structures of SR9009 (**a**) and STL1267 (**b**). **c** FRET assay demonstrating NCoR ID1 recruitment to REV-ERBα in the presence of STL1267 (λ; blue) or SR9009 (ν; light blue). Results of a REV-ERBα scintillation proximity radioligand binding assay illustrating displacement curves using either unlabeled STL1267 (**d**) or SR9009 (**e**) to displace ³H-STL1267 from the REV-ERBα LBD. **f** Cell-based two-hybrid luciferase reporter assay demonstrating the ability of either STL1267 (○; light blue) or SR9009 (□; blue) to drive recruitment of NCoR1 by REV-ERBα dose-dependently. HEK-UAS-luc reporter cells were co-transfected with NCoR-Vp16 and REV-ERBα-FL/GAL4-DBD constructs. Data are presented as mean ± SEM for **c**, **d** and **e** and mean ± SD for **f**. Each point in the biochemical and cell-based experiments represent triplicate determinations and experiments were typically repeated 3 times.

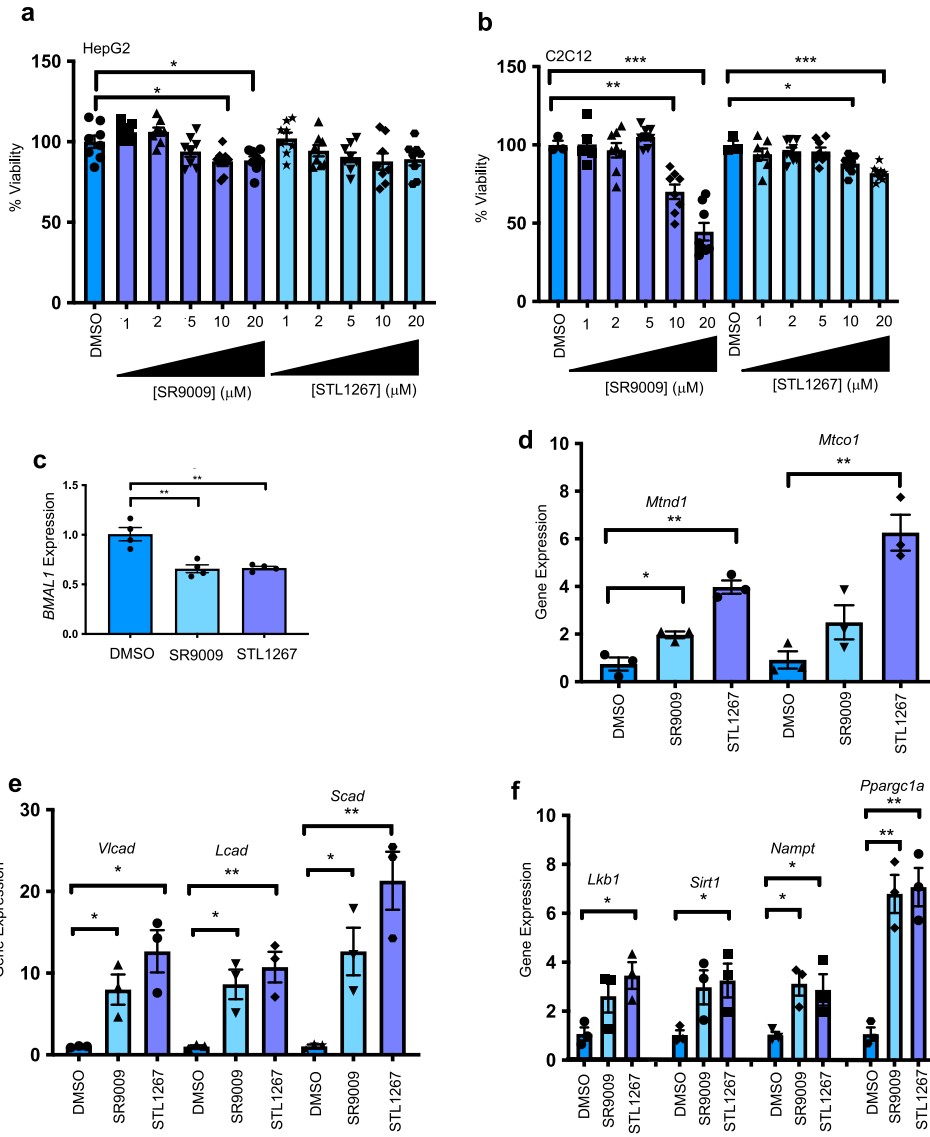

**Fig. 2 | STL1267 regulates REV-ERB target genes in cell-based assays. a** HepG2 cells treated with DMSO (blue), SR9009 (purple) or STL1267 (light blue) for 24 h (n = 3 to 8) followed by assessment of cell viability by crystal violet staining. DMSO vs. 10 μM SR9009. **b** Proliferating C2C12 cells treated with DMSO (blue) SR9009 (purple) or STL1267 (light blue) for 24 h (n = 3 to 8) followed by assessment of cell viability by crystal violet staining. (DMSO vs. 10 μM SR9009 P = 0.0003; DMSO vs. 20 μM SR9009 P < .0001; DMSO vs. 20 μM STL1267 P = 0.0481). **c** Expression of *BMAL1* in HepG2 cells in response to 24 h treatment with DMSO (blue), 5 μM STL1267 (purple) or SR9009 (light blue) (n = 4) P = 0.0010 vehicle vs. STL1267, P = 0.0009 vehicle vs. SR9009. Relative gene expression in C2C12 cells (n = 3) in response to REV-ERB agonist treatment (5 μM)

was assessed for mitochondrial complex genes (*Mt-Nd1* − DMSO vs. SR9009 P = 0.0024 and vs. STL1267 P = 0.0001; *Mt-Co1* − DMSO vs. STL1267 P = 0.0005) (**d**), fatty acid oxidation (*Vlcad* − DMSO vs. SR9009 P = 0.062 and vs. STL1267 P = 0.0074; *Lcad* − DMSO vs. SR9009 P = 0.0209 and vs. STL1267 P = 0.007; *Scad* − DMSO vs. SR9009 P = 0.0374 and vs. STL1267 P = 0.0030) (**e**), and mitochondrial function/biogenesis (*Lkb1* − DMSO vs. STL1267 P = 0.0316; *Sirt1* − DMSO vs. STL1267 P = 0.0600; *Ppargca1* − DMSO vs. SR9009 P = 0.0015 and vs. STL1267 P = 0.0012; *Nampt* − DMSO vs. SR9009 P = 0.0338 and vs. STL1267 P = 0.0542) (**f**). *P < 0.05, **P < 0.01, ***P < 0.001 and ****P < 0.0001 vs DMSO or vehicle control by one-way ANOVA followed by Dunnett's post hoc test. Data are presented as mean ± SEM. Source data are available as a Source Data file.

Using custom tritiated STL1267 we performed a scintillation proximity assay (SPA) with beads bound to the REV-ERBα LBD. The beads were saturated with radioligand, $^3$H-STL1267, and this tritiated ligand was displaced with unlabeled ligand (STL1267 or SR9009). Unlabeled STL1267 effectively displaced $^3$H-STL1267 from REV-ERBα ($K_i$ = 0.16 μM) forming a displacement curve indicating one single binding site of STL1267 to REV-ERB (Fig. 1d). SR9009 also effectively displaced $^3$H-STL1267 from REV-ERBα ($K_i$ = 0.68 μM) (Fig. 1e) suggesting overlap in their binding sites. Due to the optical properties of heme, we were unable to utilize this ligand in the radioligand binding assay.

We next assessed the activity of STL1267 in a cell-based assay. We utilized a chimeric construct expressing full length human REV-ERBα (REV-ERBα-FL) fused to the DNA-binding domain (DBD) of yeast transcription factor GAL4. HEK293 cells stably expressing a GAL4 responsive luciferase reporter were co-transfected with REV-ERBα GAL4-DBD and NCoR-Vp16. This format of the assay essentially functions as a mammalian 2-hybrid assay and detects REV-ERB agonists as activators of transcription effectively enhancing the sensitivity of such an assay from the typical one hybrid system that detects enhancement of transcriptional repression. We again compared STL1267 to SR9009 in this assay, and as illustrated in Fig. 1f, STL1267 displayed enhanced

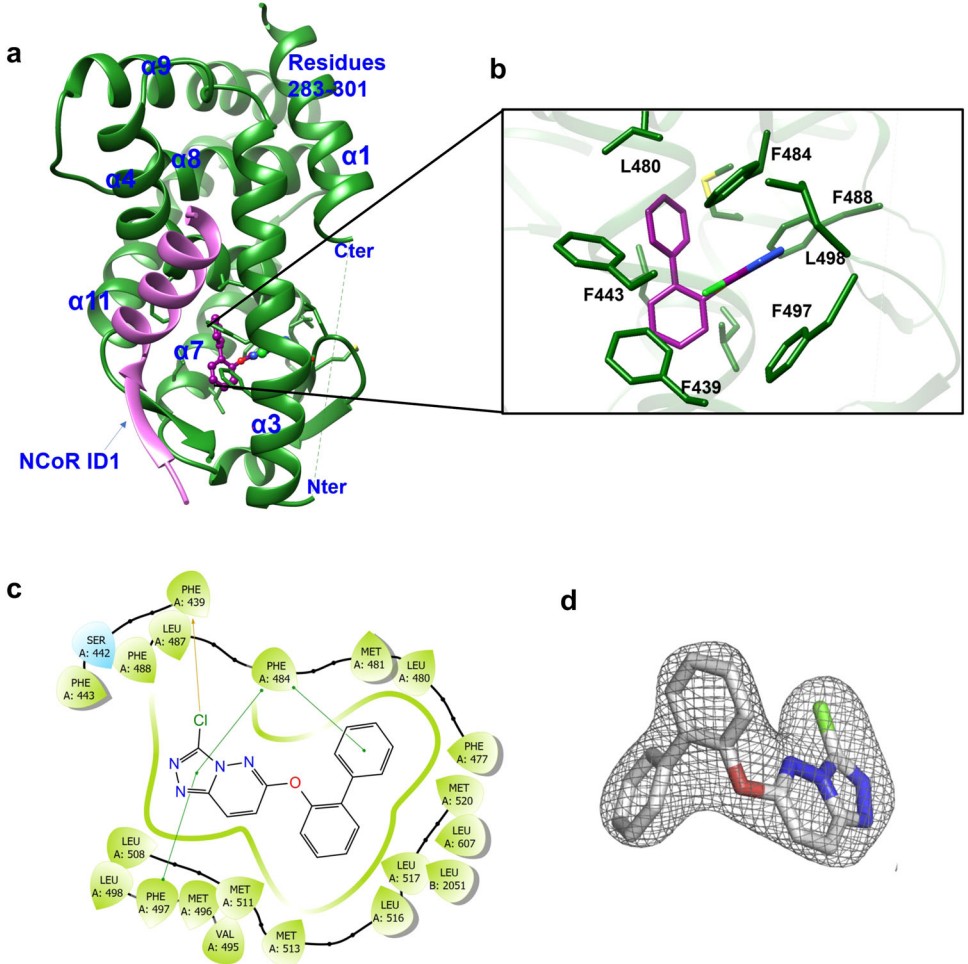

**Fig. 3 | Crystal structure of STL1267 bound to the ligand binding domain of REV-ERBα. a** Crystal structure of STL1267 bound to the ligand binding domain of REV-ERBα with NCoR ID1. REV-ERBα is illustrated in green ribbons and NCoR ID1 in purple ribbons. The ligand is shown as ball and stick representation. Common structural components (α helices) within the nuclear receptor LBD are designated. **b** Close-up view of STL1267 binding within the ligand binding pocket of REV-ERBα. The ligand is illustrated as purple stick representation and the protein amino side chains are demonstrated by green stick representation. Hydrophobic amino acids stabilize STL1267 in the ligand binding pocket of REV-ERBα. **c** 2D diagram illustrating STL1267 interactions with amino acid residues within the ligand binding pocket of REV-ERBα. The label A corresponds to the REV-ERB protein residues while label B corresponds to the NCOR residues. (d) $2f_o - f_c$ electron density map around ST1267 contoured at 2σ.

potency and efficacy relative to SR9009. STL1267 displayed an $EC_{50}$ of 1.8 μM vs. 4.7 μM for SR9009. The maximal efficacy for recruitment of NCoR-VP16 was ~8-fold greater for STL1267 vs. SR9009.

We assessed the specificity of STL1267 using both in house nuclear receptor Gal4-NR reporter assays and the NIMH Psychoactive Drug Screening Program (PDSP) assays. We assessed the specificity of STL1267 in the PDSP's radioligand displacement assays, screening for binding to off-target GPCRs, ion channels, transporters, and other drug targets (Table S1). Of the 43 targets assessed in this panel, we observed no detectable binding in 41 of them. We detected weak binding to the serotonin transporter ($K_i \sim 2-4$ μM). The kappa opioid receptor also displayed activity in the binding assay with a $K_i$ of 0.72 μM, but this is clearly not as high affinity as STL1267 has for REV-ERB. Additional assessment in serotonin transporter and kappa opioid receptor activity in a functional assay may be warranted to determine absolute specificity. We did not detect activity of STL1267 against a range of NRs including AR, GR, MR, PR, ERα, ERβ, TRα, LXRα, RARα, RXRα, FXR, and VDR (Table S1) using a cotransfection assay as previously described[22–24].

Together, these data indicate direct binding of STL1267 to REV-ERBα, followed by NCoR recruitment. STL1267 binds more potently to REV-ERBα than SR9009 and appears to recruit corepressor more

efficaciously. In addition, the specificity profile of STL1267 is very favorable with no activity at a range of other ligand-regulated NRs and minimal binding at other classes of receptors/transporters.

## STL1267 regulates expression of REV-ERB target genes

A relatively recent study indicated that SR9009 displays general toxicity in cell-based assays[17]. Although we and others have not observed this degree of toxicity[2–4,6–8,25,26], we sought to compare the effect of SR9009 to STL1267 on cell viability in two commonly used cell lines. Cellular viability was tested in two cell types, human hepatocarcinoma HepG2 cells and proliferating C2C12 mouse myoblast cells. In HepG2 cells, SR9009 had minimal effect on cell viability (~10% reduction at 10 and 20 μM) while STL1267 showed no adverse effects on cell viability up to the maximum dose examined (20 μM) (Fig. 2a). Proliferating C2C12 cells were considerably more sensitive with SR9009 displaying much greater reduction in cellular viability (~25% reduction at 10 μM and ~50% at 20 μM) (Fig. 2b). In these cells, STL1267 displayed a much more favorable profile than SR9009 relative to reduction in viability (~10% reduction at 10 μM and ~20% reduction at 20 μM). It is important to note that in both cell lines with either drug, doses of 5 μM and below displayed no indication of reduction in cellular viability.

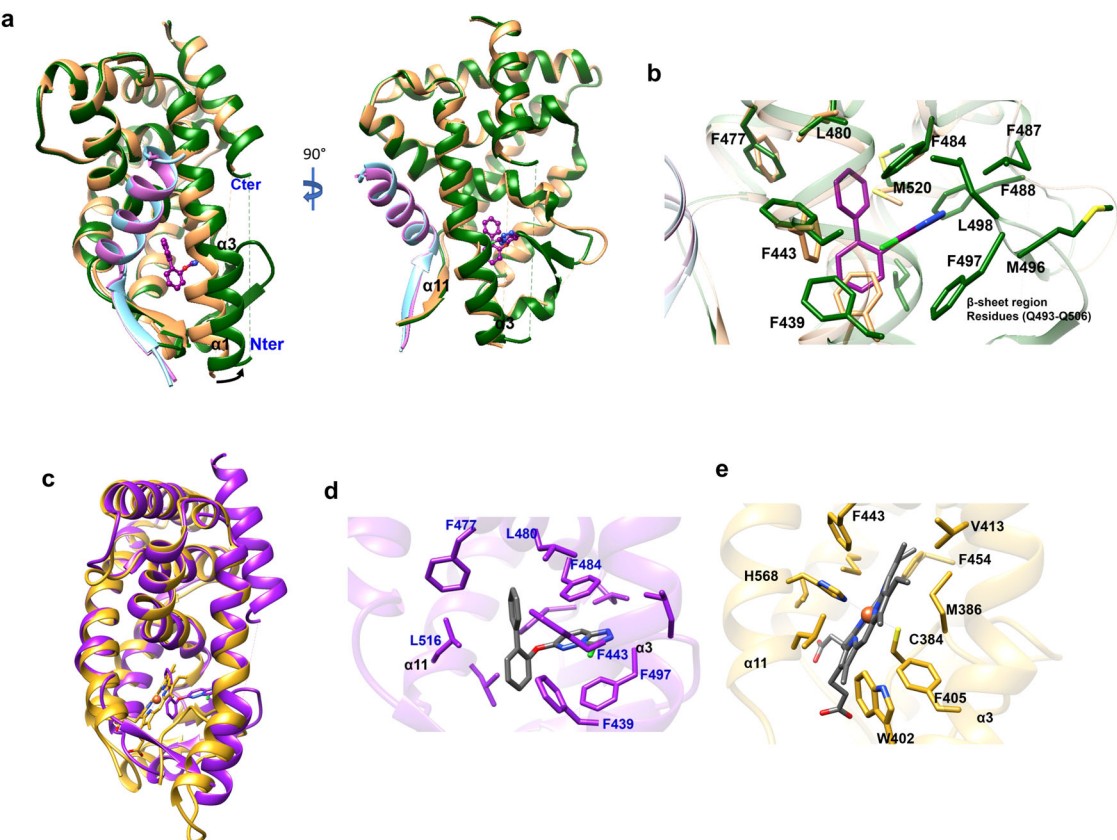

**Fig. 4 | Schematic illustrating effects of STL1267 binding and comparison to apo REV-ERBα and heme bound REV-ERBβ. a** Overlay of the REV-ERBα/NCoR ID1/ STL1267 (protein is shown as green ribbons, NCoR peptide as blue ribbons and ligand carbons are shown in purple) and REV-ERBα/NCoR ID1 (protein is shown as orange ribbons, NCoR peptide as pink ribbons, PDB:3N00). **b** Detailed view of the superimposed ligand binding pockets of both structures. Orange amino acid residues corresponds to the apo REV-ERBα, green amino acid residues correspond to the STL1267/REV-ERBα. The ligand is illustrated by purple carbons. **c** Overlay of the REV-ERBα/STL1267 (purple) and REV-ERBβ/heme (yellow). **d** Detailed view of STL1267 within the REV-ERBα LBD. Ligand is colored gray. **e** Detailed view of heme within the REV-ERBβ LBD (heme is colored gray and the orange sphere represents iron).

We next determined whether STL1267 can regulate characterized REV-ERB modulated genes in cell lines expressing REV-ERB naturally. Five micromolar doses of both drugs were used in cell-based experiments from this point on. REV-ERB is an established component of the core molecular clock repressing the expression of core clock gene *Bmal1*[6,27] as well as others[28,29]. Using a human hepatocarcinoma cell line, HepG2, STL1267 attenuated the expression of REV-ERB target gene and circadian modulator, *Bmal1*. In this assay, STL1267 was more efficacious than SR9009 in repressing *Bmal1* gene expression (Fig. 2c). In skeletal muscle, REV-ERB agonism increases the expression of a wide variety of genes important in mitochondrial function[5]. When proliferating mouse myoblast C2C12 cells were treated with either REV-ERB agonist, expression of mitochondrial complex (mitochondrially encoded NADH:ubiquinone oxidoreductase core subunit 1 - *Mtnd1*, mitochondrially encoded cytochrome C oxidase I -*Mtco1*), fatty acid oxidation (very long chain acyl-CoA dehydrogenase-*Vlcad*, long-chain acyl-CoA dehydrogenase-*Lcad*, Short-chain acyl-CoA dehydrogenase-*Scad*), and mitochondrial function/biogenesis (liver kinase B1-*Lkb1*, sirtuin 1-*Sirt1*, Nicotinamide phosphoribosyltransferase-*Nampt*, peroxisome proliferator-activated receptor coactivator 1-*Ppargc1a*) genes were upregulated (Fig. 2d–f). All of these genes were demonstrated to be responsive to REV-ERB previously[5]. STL1267 was more efficacious than SR9009 in upregulating of several of these gene targets (Fig. 2c–f). These data demonstrate that STL1267 modulates REV-ERB target genes in cell-based models and displays similar activity as SR9009 with the exception that it appears to be more efficacious in terms of maximal gene regulation in many cases.

## Characterization of STL1267 in vivo

To test the efficacy of STL1267 in vivo, we began by assessing tissue exposure following i.p. administration. We administered vehicle or 50 mg/kg of STL1267 via intraperitoneal injection to C57Bl/6 J mice. The mice were sacrificed at four subsequent time points over the next 12 h. The plasma half-life of STL1267 was ~1.6 h, and the compound was localized in all the tissues collected (brain, plasma, liver, skeletal muscle, white adipose tissue) (Supplementary Fig. 2a). Brain levels were similar to plasma levels indicating that STL1267 successfully crosses the blood brain barrier (Supplementary Fig. 2a). Given that we had good exposure following i.p. administration, we also assessed *Bmal1* expression in liver at 12 h post administration and observed that STL1267 effectively suppressed *Bmal1* expression consistent with its function as a REV-ERB agonist (Supplementary Fig. 2b). Together, these data suggest that STL1267 is a valuable tool compound for studying REV-ERB function both in vitro and in vivo.

## Crystal structure of a synthetic agonist bound to the REV-ERBα LBD

Although the crystal structures of the apoREV-ERBα[30] and -β[31] LBDs have been determined as well as the structure of natural porphyrins (heme[14,15] or cobalt protophorphyrin[32]) bound to REV-ERBβ, no structures of non-porphyrin, synthetic ligands bound to either REV-ERBs have been described. Attempts to determine the structure of SR9009 bound to REV-ERB had failed and with the higher affinity STL1267 in hand, we sought to determine if this would aid in determination of a structure.

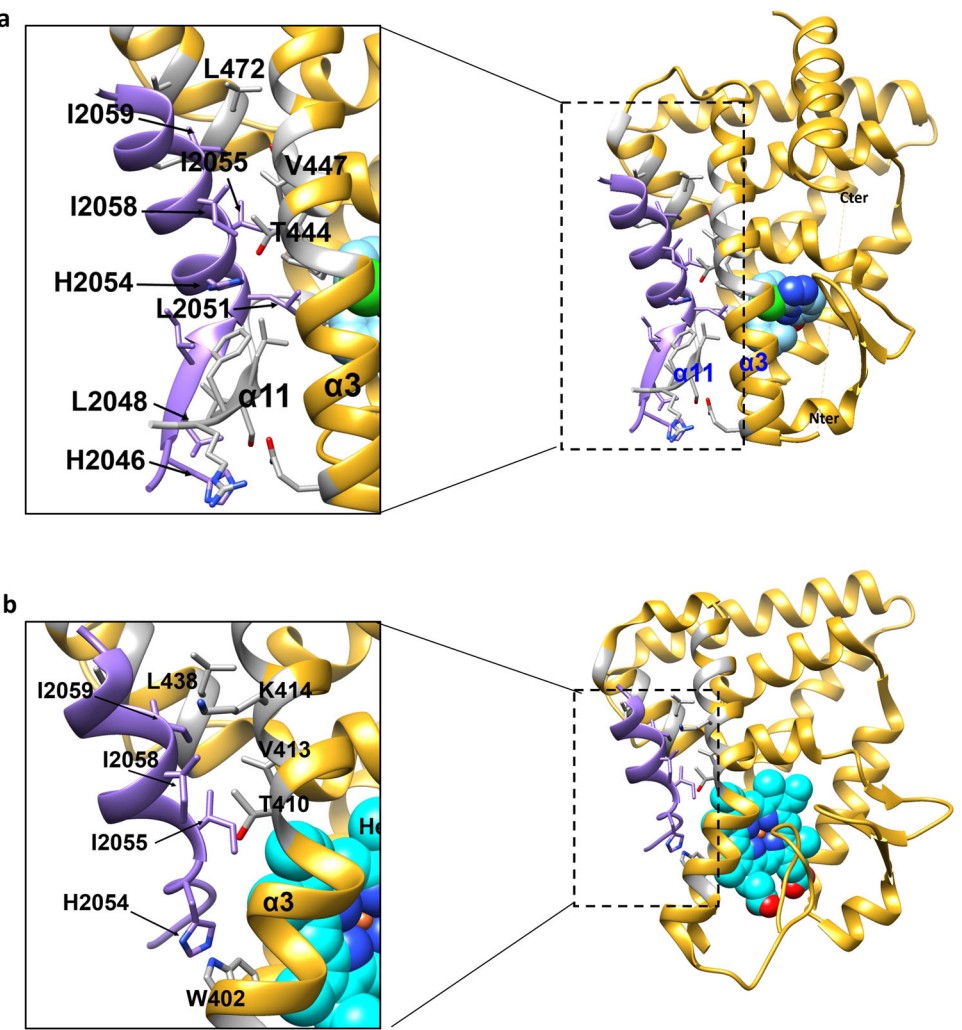

**Fig. 5 | Distinct conformations of NCoR1 ID1 peptide binding to REV-ERB induced by STL1267 vs heme. a** Structure of STL1267/REV-ERBα LBD (yellow) bound to NCoR1 ID1 peptide (purple). Regions of the receptor interacting with NCoR are represented in gray. STL1267 is represented in space filled format.

**b** Structure of heme/REV-ERBβ LBD (yellow) bound to NCoR1 ID1 peptide (purple) (PDB: 6WMQ). Regions of the receptor interacting with NCoR are represented in gray. Heme is represented in space filled format.

We determined the crystal structure of a ternary complex containing the REV-ERBα LBD bound to the corepressor NCoR ID1 CoRNR box peptide and the agonist STL1267 (Table S3). Except for helix 12, which is naturally lacking in REV-ERBα, the overall three-dimensional structure of REV-ERBα displays the canonical architecture of nuclear receptor LBD with a three-layer α-helical sandwich and two stranded β-sheets. The C-terminal of helix 11 assumes a well-defined extended β-strand conformation allowing for the formation of an antiparallel β-sheet interactions with N-terminal residues of NCoR ID1. Besides the β-sheet interaction, NCoR ID1 contains a carboxy-terminal four-turn helix (Fig. 3a) that docks into the coregulator binding cleft of REV-ERBα LBD, known as the AF-2 surface. The ligand resides in a predominantly hydrophobic pocket where it makes primarily hydrophobic interactions with the receptor. The triazolo-pyridazine group of STL1267 efficiently fills the narrow portion of the hydrophobic pocket and forms π-π stacking interactions with Phe484 and Phe497 (Fig. 3b, c). We compared the structure of the REV-ERBα/NCoR ID1/STL1267 complex with the previously solved structures of the apo REV-ERBα/NCoR ID1 (PDB: 3N00)[30] and REV-ERBβ/Heme (PDB: 3CQV)[14] complexes. The global receptor conformation of STL1267-bound REV-ERBα LBD was very similar to that of the apo REV-ERBα (overall Cα RMSD of 1.07 Å) (Fig. 4a). However,

superposition of both structures indicates several differences in the ligand binding pocket to accommodate ligand binding. First, binding of STL1267 stabilized the β-sheet region (residues Gln493-Gln506) through hydrophobic interactions with Phe497 and Leu498, which is disordered in the apo REV-ERBα structure (Fig. 4b). Outward displacement of helix 3 and change of conformation of amino acid Met529, Phe439 and Phe443 were observed, which provided space for ligand binding (Fig. 4b).

The binding mode of STL1267 differs substantially from heme binding in the REV-ERB LBP[14]. Although the STL1267 structure is crystallized with REV-ERBα and the heme structures are crystallized with REV-ERBβ, both REV-ERBα and REV-ERBβ have high degree of similarity and the residues within REV-ERBβ demonstrated to interact with heme are conserved in both isotype suggesting that heme-bound REV-ERBα will have similar structural features to REV-ERBβ bound with heme (Supplementary Fig. 3). Heme binds in a solvent exposed region in the LBD near helix 11 where several water molecules observed in the LBP make hydrogen bonds with the carboxylic groups of heme while its central iron molecule is coordinated by residues His586 and Cys384 of REV-ERBβ (Fig. 4c, e). There is minimal overlap between STL1267 and heme in terms of the occupied space in the LBP. The binding of STL1267 is quite distinct from that of heme and where heme binds proximal to helix 11, STL1267 binds distal to helix 11 and is deeply buried within a

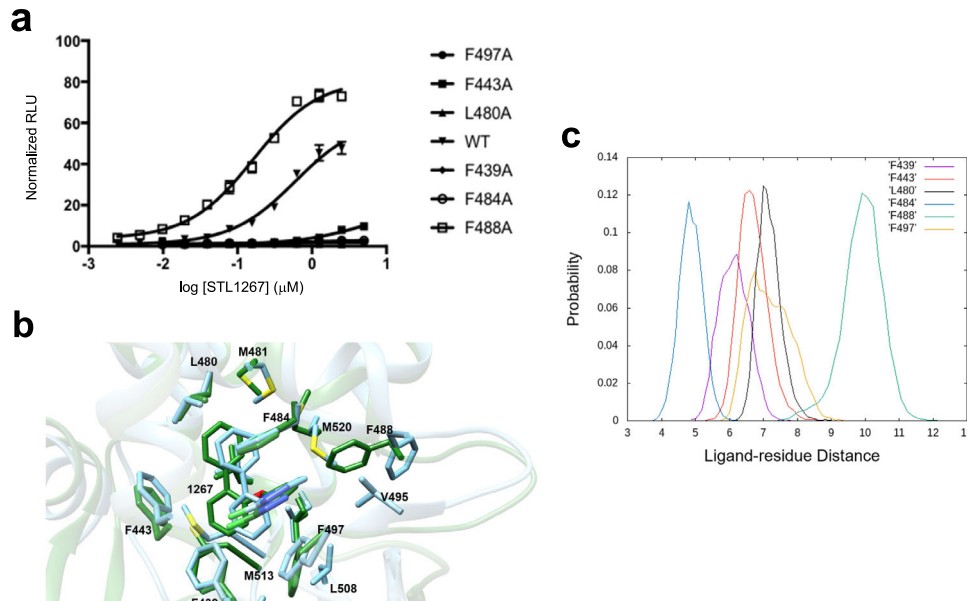

**Fig. 6 | Mutational analysis of amino acid residues within the ligand binding pocket of REV-ERBα. a** Results of a two-hybrid reporter assay (Gal4-FL REV-ERBα/ NCoR1-VP16) assay illustrating the effects of mutation of amino acid residues that interact with STL1267 within the ligand binding pocket. (WT ▼, F497A ●, F443A ■, L480A ▲, F439A ◆, F484A ○, F488A □). Each point in the biochemical and cell-based experiments represent triplicate determinations and experiments were

typically repeated three times. Data are presented as mean ± SEM. **b** Overlay of the ligand binding pockets of REV-ERBα/STL1267/NCoR ID1 complex before (green) and after (blue) the MD simulations. **c** Distance probability distribution for STL1267 and selected amino acid residues (F497A yellow, F443A red, L480A black, F439A purple, F484A blue, F488A green).

hydrophobic pocket surrounded by helices 3 and 5 and the two β-sheets strands (Fig. 4c–e). Although the corepressor binding surface of both heme- and STL-1267 bound REV-ERB LBD was preserved with the α-helical region of the ID1 peptides containing the I/LxxI/LI corepressor NR (CoRNR) box motif binding effectively, major distinctions were observed in the mechanism of corepressor binding between the two ligand-bound receptors. STL1267 binding induces an antiparallel β-sheet formation between the amino-terminus of the NCoR ID1 CoRNR box peptide and a β-strand extension of helix 11 (Fig. 5a), which was not observed in the heme-bound receptor (Fig. 5b)[15]. The more proximal binding of heme to this region of the receptor that interacts with the CoRNR box peptide (relative to STL1267 binding) leads to heme itself sterically hindering formation of the anti-parallel β-sheet that is observed in the STL1267-bound receptor.

Given the distinct interactions that the synthetic ligand displays in the LBD relative to heme, we sought to confirm the binding mode by mutating key amino acid residues that interact directly with STL1267 to alanine (i.e., Phe497, Phe443, Phe480, Phe439, Phe484, and Phe488 (Fig. 4d). We assessed the impact of these mutations in a cell-based cotransfection model where we measured the ability of the REV-ERBα to recruit a fragment of the NCoR1 corepressor. Plasmids directing the expression of WT or mutant Gal4DBD- full length REV-ERBα fusion and NCoR1-VP16 fusion protein were transfected into HEK293 cells with a Gal4UAS luciferase reporter stably integrated. As shown in Fig. 6a, STL1267 effectively induces interaction between REV-ERBα and NCoR and mutation of nearly all the amino acid residues that were hypothesized to be critical for STL1267 binding (F497A, F443A, L480A, F439A, and F484A) resulted in loss or severe reduction in activity. Surprisingly, one mutation (F488A) led to an impressive improvement in potency and efficacy in REV-ERBα's NCoR recruitment activity (Fig. 6a). The removal of the bulkier Phe488 sidechain may allow for the ligand binding pocket to wrap around the ligand more tightly, which may explain the higher activity of this mutant REV-ERB. These data indicate that the residues identified in the crystal structure as critical for STL1267 binding are indeed critical for STL1267 regulated activity of the receptor.

We performed molecular dynamics (MD) simulations of wild-type REV-ERBα and the mutant F488A bound with STL1267 and NCOR ID1 peptide to develop a hypothesis as to why the F488A mutation led to a gain of function. In the wild-type simulations, STL1267 binding to the receptor structure was stable with the biphenyl ring moving slightly deeper inside the LBP (Fig. 6b). STL1267 maintained interactions with Phe497, Phe443, Phe480, and Phe439 where the calculated distances probabilities between STL1267 and interacting amino acid side chains ranged between 5 and 7 Å (Fig. 6c). STL1267 lost its interaction with the phenyl group of Phe488 in accordance with the mutagenesis experiments where the STL1267 potency increased in the Phe488A mutant. The distance probability between STL1267 and Phe488 in the MD simulations was 10 Å (Fig. 6c). The interaction of STL1267 and Phe488 is lost during the simulations because the phenyl group of Phe488 flipped outside of the LBP and formed hydrophobic interactions with side chains of amino acid residues Phe521, Val490 and Leu505 (Supplementary Fig. 4). The loss of Phe488 interaction with STL1267 was compensated by STL1267 interactions with the methyl groups of Met520 and Val495 side chains, which occupied the vacant space created by the side chain of Phe488 flipping out of the LBP (Fig. 6b). In the molecular dynamics simulations of the F488A mutant, STL1267 maintained similar pattern of interactions with Phe497, Phe443, Phe480, Phe439, Met520, and Val495 (Supplementary Fig. 5). We also utilized Molecular Mechanics Generalized Born Surface Area (MM/ GBSA)[33] to qualitatively estimate the binding free energies of STL1267 to the wild-type and mutant REV-ERBα receptor (Supplementary Table 4). In the case of the F488A mutant, both the enthalpy (ΔH) and entropy (ΔS) contribution of STL1267 to the total binding free energy, ΔG, were more favorable (ΔG value is −24.4 Kcal/mol for the mutant versus a ΔG value of −20.8 Kcal/mol for the wild type). Thus, the simulations suggest that the aromatic side chain of Phe488 is not favorable for ligand binding at this contact point of the ligand, rather the methyl groups of nearby amino acid side chains were preferred for ligand interaction. Therefore, it is not surprising that mutation of Phe488 to alanine resulted in gain of STL1267 potency as alanine has smaller size side chain providing additional vacant space for the

nearby amino acid side chains of M520 and Val495 to interact freely with the ligand.

Another interesting insight revealed by the MD simulations of REV-ERBα/STL1267/NCoR ID1 complex is the increased flexibility of the β-sheet regions of NCoR (amino-terminal) and REV-ERBα (carboxy-terminal) as observed in the MD simulations trajectory (Supplementary Movie 1 and Supplementary Data 1 and 2) and indicated by increased fluctuations (RMSF) of the β-sheets region (RMSF~ 6–9 Å) while the N-CoR α-helical region was dynamically more stable (RMSF < 3 Å) (Supplementary Fig. 7). These results infer that the NCoR α-helical region is more important for interaction with REV-ERBα. This is in accordance with previously reported structural and mutagenesis data where alanine mutations of the NCoR α-helical residues, Ile2055, Ile2058, Ile2059 (ICQII motif) abolished the interaction with REV-ERBα, whereas mutations in the extended N-terminal β-sheet (LADH extension), residues His2046, Leu2048, Leu2051, and His2054 had little effect[30].

## Discussion

The porphyrin, heme, was identified as a physiologically relevant ligand for REV-ERBα[11,12] and REV-ERBβ[11] in 2007. Prior to this discovery, the REV-ERBs were believed to be constitutive repressors of expression of their target genes; however, these constitutive effects are likely due to the constitutive availability of heme rather than ligand-independent activity of REV-ERB. Mutation of amino acid residues within the LBP of REV-ERB that are responsible for coordinating the iron ion within the porphyrin ring eliminate the ability of REV-ERB to bind to heme and render the receptor unable to modulate gene transcription[11,12]. The observation that the REV-ERBs are ligand-dependent transcriptional repressors led to significant efforts to develop agonists and antagonists that could be used as chemical tools to modulate the activity of these receptors in cell- and animal-based models given that the REV-ERBs regulate myriad physiological pathways involved in many disease processes[13,34–36]. These efforts led to the development of a number of compounds that are now used as standard pharmacological chemical tools targeting REV-ERBs including the agonists GSK4112[16,37], SR9009[6], and the antagonist SR8278[38]. Although these chemical tools, in combination with REV-ERB genetic loss of function models, have been effective in determining the potential therapeutic value of synthetic REV-ERB ligands improvements in the pharmacodynamic and pharmacokinetic properties of such ligands has been slow to progress. Any efforts to develop synthetic ligands based on the natural porphyrin ligand for REV-ERB have been avoided due to heme binding to a vast array of proteins and the prediction that REV-ERB specificity with such a chemical scaffold will be insurmountable.

Crystal structures of the apoREV-ERBα[30] and -β[31] LBDs have been determined, but the only structures of REV-ERB LBD bound to ligands are of porphyrins ((heme[14,15] (agonist) or cobalt protophorphyrin[32] (antagonist)). Given the expected unique nature of porphyrins binding within the LBD of REV-ERB relative to more drug-like synthetic ligands, such as histidine/cysteine residues coordinating metal ions, these crystal structures have not been very informative about how non-porphyrin ligands may bind and have not been useful from the perspective of rational drug design. Our assessment of the activity of a high-affinity non-porphyrin REV-ERB agonist, STL1267, enabled our efforts to develop a radioligand binding assay for REV-ERB as well as to determine the crystal structure of a non-porphyrin agonist bound to the LBD of REV-ERBα. In addition to being considerably more potent (STL1267 REV-ERBα $K_i = 0.16\,\mu M$ vs. $0.68\,\mu M$ for SR9009), STL1267 was also much more efficacious than SR9009 in biochemical and cell-based assays detecting recruitment of NCoR. This appeared to translate into greater maximal efficacy of STL1267 regulation of target REV-ERB genes in many cases.

Although the STL1267/REV-ERBα LBD/NCoR1 ID1 peptide crystal structure showed similar global structure to other REV-ERB LBD

structures, there were two very important distinctions. Firstly, the mechanism of how STL1267 bound to the LBD was very distinct from that described for the porphyrin ligands. Although heme and STL1267 displayed some overlap in the regions occupied within the LBP of REV-ERBα, heme occupied a region much more proximal to helix 11 and the corepressor binding surface. In fact, heme was partially solvent exposed in this region. Water molecules occupied the LBP of the heme bound LBD where they interacted with the carboxylic acid moieties of heme. The central iron ion was also coordinated by a key histidine and a key cysteine residue. In contrast, STL1267 bound to the LBD in a deeply buried hydrophobic pocket much more distal to helix 11 where the interactions with the residues within the LBP were hydrophobic in nature. Even with these distinctions in their modes of binding, both agonists led to effective binding of the NCoR CoRNR box ID1 peptide. However, this leads to the second distinction in the structures. Part of the heme molecule actually extends out of the point of entry into the LBP of REV-ERBα, which leads to an altered structure of the NCoR1 peptide. Although the core α-helical I/LxxI/LI component of the peptide was identical in both structures and docked with the "coactivator" binding cleft of the LBD, the amino-terminal extension of the CoRNR box peptide formed an antiparallel β-sheet with carboxy-terminal component of helix 11 in the STL1267-bound structure only. In the heme-bound structure, the ligand heme extended out of the ligand binding pocket interfering with the ability of helix 11 to form such an anti-parallel β-sheet structural motif and the residues that formed this motif in the NCoR peptide were not resolved in the structure. This suggests that STL1267-bound receptor leads to a distinct complex mode of interaction with the corepressor than heme-bound receptor. Although it is currently unclear, such additional interactions between REV-ERB and corepressor induced specifically by STL1267 may yield unique pharmacological actions (e.g. altered efficacy or modulator-like profile).

We compared the activity of STL1267 to SR9009 and given a recent study[17] indicating that SR9009's toxicity (independent of REV-ERB activity) may preclude its use as a chemical tool. We treated either C2C12 cells or HepG2 cells with increasing concentrations of the two drugs and examined toxicity by crystal violet staining. Dierickx et al. reported severe SR9009 toxicity in mouse embryonic stem cells (ESCs) independent of REV-ERBα/β expression as well as in a range of other cell types[17]. The authors also observe decreases in mitochondrial respiration with SR9009 in ESCs, which aligns with the ATP-dependent endpoint of their cellular viability assays[17]. These observations are in contrast to those where SR9009 increased mitochondrial mass and membrane potential in C2C12 cells[5] ($5\,\mu M$) and enhanced mitochondrial respiration in retinal pigment epithelial cells[39] ($1\,\mu M$). Thus, these effects appear to be very cell type specific as well as dependent on the dose of SR9009 used as we observed in C2C12 vs. HepG2 cells (Fig. 2a, b). In terms of reduction in cell viability, C2C12 cells were much more sensitive to both drugs relative to HepG2 cells, but in general, STL1267 had much reduced effect on cellular viability in general. The availability of a REV-ERB agonist with greater potency than SR9009 with a distinct chemical scaffold provides an important tool in REV-ERB pharmacology to allow for examination of potential REV-ERB specificity issues. One would expect that off-target effects would not necessarily be shared between two distinct chemical scaffolds and comparison of the effects of these drugs may help to define which are REV-ERB driven. The lack of complete REV-ERBα/REV-ERBβ KO mouse model to evaluate SR9009 has led to some controversy as to the specificity of this compound. In many cases, SR9009 gain-of-function has been compared to REV-ERBα genetic loss-of-function (KO) or gain-of-function phenotypes (overexpression) to correlate the potential role of REV-ERB as well as potential therapeutic value of pharmacological targeting this receptor[2–6,24–27,40–43]. In

other cases, SR9009's efficacy in WT mice/cells was compared to efficacy in the REV-ERBα KO mouse model or cellular REV-ERBα KD to determine if there is a degree of efficacy lost[1,2,44–47]. Another issue that has arisen is the use of floxed allele KO mice as models to examine receptor specificity of drugs such as SR9009. In many cases these models produce hypomorphs that represent significantly reduced expression of the receptor, but not complete loss. For example, when we assessed the RNA-seq data obtained from hepatocytes (treated with SR9009 or vehicle) from *Rev-erbα/β* <sup>fl/fl fl/fl</sup> mice *treated* in vivo with AAV-Cre (tail vein injection)[17] we observe that 39% of genes identified as REV-ERB regulated genes based on comparison of WT vs Rev-*erbα/β* <sup>fl/fl fl/fl</sup> DKO hepatocytes were still regulated by SR9009 in the *Rev-erbα/β* <sup>fl/fl,fl/fl</sup> DKO hepatocytes[48]. Although one might expect to observe some degree of overlap in SR9009-regulated genes that are not REV-ERB-dependent with those that are REV-ERB-dependent, a 39% overlap is more consistent with a hypomorph phenotype. Yet another issue to consider when using this type of gene expression analysis as indicator specificity of a drug (particularly for a tool compound), one must consider that the agents are xenobiotics and pharmacokinetic parameters must be considered including genes of the xenobiotic response which will likely be regulated to a significant degree independent of the intended target. The addition of STL1267 to the chemical tools that can be used to probe REV-ERB function will aid in the assessment of specificity as well.

## Methods

### HEK293 and HEK293 UAS-luc

Prior to use in co-transfection assays, the human embryonic kidney (HEK293) cell line (ATCC, CRL-1573) was cultured in 10% FBS DMEM at 37 °C in a humidified atmosphere of 5% $CO_2$. HEK UAS-luc cells contain a plasmid that is resistant G418 (Geneticin). This antibiotic was added to the media of these cells to preserve the HEK UAS-luc cell line.

### Cell culture

The human hepatocellular carcinoma cell line, HepG2 cells (ATCC HB-8065), were cultured in 10% FBS 1% L-Glu MEM at 37 °C in a humidified atmosphere of 5% $CO_2$. Prior to RNA extraction cells were treated with either DMSO or REV-ERB agonist for 24 h. The C2C12 proliferating mouse myoblast cells (ATCC, CRL-1772) were cultured in 10% FBS DMEM at 37 °C in a humidified atmosphere of 5% $CO_2$. Prior to RNA extraction cells were treated with either DMSO or REV-ERB agonist for 24 h. Prior to crystal violet cytotoxicity assay, cells were treated with either DMSO or REV-ERB agonist for 48 h.

### Radioligand binding assays

For radioligand displacement assays, an 11-point serial dilution of cold REV-ERB agonist, STL1267 or SR9009, was prepared in buffer (20 mM potassium phosphate pH 7.4, 50 mM KCl, 0.005% TWEEN20, 1 mM DTT). On top of the drug, a solution of protein/radioligand mix was added to the assay plate. The final concentration of the protein/radioligand mix in the plate were as follows: 80 nM $^3$H-1267/96.9 µCi and 120 nM (0.57 µg/well) "heme" REV-ERBα. YSI Copper Ni SPA beads were added to a final concentration of 240 µg/well. The contents of the plate were mixed on a shaker (400 rpm) for 20 min and then allowed to sit for 10 min prior to read-out of scintillant emission. Data were normalized by taking the average of the DMSO value per plate (the maximum binding) and dividing all the test values from the same plate by this DMSO average. The normalized values were transformed using X = Log(X) in Prism. The IC50 was derived from the transformed values using "One site – Fit logIC50" in the "Binding – Competitive" folder. The Ki was derived from the transformed values using "One site – FitKi" in the "Binding – Competitive" folder. Radioligand binding assays intended to determine 1267 specificity were carried out by the NIMH

Psychoactive Drug Screening Program in accordance with the PDSP protocol (https://pdsp.unc.edu/pdspweb/content/PDSP%20Protocols%20II%202013-03-28.pdf).

### FRET

A serial dilution of REV-ERB agonist, STL1267 or SR9009, was prepared in buffer (0.1% Triton, 5 mM DTT, PBS). Histidine tagged REV-ERBα/β complexed with heme was added to each well to a final concentration of 4 nM. NCoR ID1 fluorescein (ref#PV4622, Batch# 1797468C) was added to each well to a final concentration of 150 nM, and terbium anti-histidine (ref#PV5895, batch 1730002B) was added to a final concentration of 1 nM. Mixture was incubated at room temperature on an orbital shaker for 1 h at 200 rpm. The lanthascreen assay protocol (Dual PMT, excitation 340 nm, dual emission 495/520, Gain 110/110) was used to read the plate.

### Co-transfection assays

The NCoR-Vp16 plasmid and REV-ERBα/β-FL GAL4-DBD wild-type or point mutation plasmids were incubated with the transfection agent, Lipofectamine 2000. GFP plasmid was also in the transfection mix to confirm transfection efficiency. HEK cells were plated in 10 cm dishes for western blot analysis, and HEK UAS-luc cells were plated in 96 well plates for luciferase assay analysis. Prior to cellular adherence, transfection mix was added. The cells and transfection mix were incubated at 37 °C in a humidified atmosphere of 5% $CO_2$. The transfection incubation lasted overnight for luciferase assays and 48 h for protein isolation/western blot analysis. For luciferase assays, REV-ERB agonists were added in an 11-point serial dilution the day following overnight transfection. Cells were incubated with the compound for 24 h, prior to luciferin addition and luminescence read out.

### Crystal violet cytotoxicity

Proliferating C2C12 and human peripheral blood monocytes were treated for 48 h with indicated concentrations of REV-ERB agonist or vehicle (DMSO). Media was removed, and cells were incubated with 0.5% crystal violet solution for 20 min at room temperature with mild agitation. Cells were washed four times with tap water before being air drying overnight. Cells were incubated in methanol for 20 min with mild agitation. Absorbance was read at 570 nm, and readings were normalized to vehicle (DMSO) treated cells.

### RNA extraction

Mouse tissue was homogenized using a bullet blender. DNA/protein was removed by phase separation using BCP/chloroform. The supernatant (RNA containing fraction) was collected, and RNA was precipitated using isopropanol. The RNA pellets were washed with 75% ethanol and resolubilized in RNAse free water. The purity of the RNA was tested by agarose gel prior to cDNA synthesis. Total RNA was harvested from HepG2 and C2C12 using the RNeasy mini kit according to manufacturer's instructions (Qiagen).

### Quantitative PCR

Isolated RNA was reverse transcribed into cDNA using the qScript™cDNA Synthesis kit according to the manufacturer's instructions (QuantaBio). Quantative PCR (qPCR) was performed on a QuantStudio™ 6 Flex or QuantStudio™ 7 Flex (Life Technologies) using SYBR™ Select Master Mix (Applied BioSystems) with select primers (Table S2). Gene expression was normalized to *Ppia* for all data sets. Oligonucleotide sequences are provided in Supplementary Table 2.

### Mouse studies

All in vivo studies were performed using the ARRIVE guidelines and approved by the Washington University IACUC. C57Bl/6J mice (male; 6–8 weeks of age) maintained at 12:12 h light;dark, 22 °C, 40–60%

humidity. Mice were injected IP with 50 mg/kg STL1267 ($n = 3-4$ per group) or vehicle (10% DMSO, 12% TWEEN80, PBS) ($n = 3-4$ per group). Mice were weighed immediately following the injection and were weighed again immediately after sacrifice. Mice were fasted started at the time of injection. For tissue collection, three STL1267-injected mice were sacrificed at 1 h post-injection, and three more were sacrificed at 3 h post-injection. At 8 h and 12 h post-injection, six mice from each treatment group were taken down and tissue was collected. The tissues collected at all four time points include plasma, white adipose tissue, brain, liver, and skeletal muscle (quadriceps). RNA was extracted from liver samples at $T = 12$ h post-injection for analysis of REV-ERB target genes. Tissue samples were sent to Charles River Laboratories for LC-MS/MS detection of STL1267. For LC-MS/MS analysis, tissue samples were homogenized using bead beater in acetonitrile:water (3:1) at 250 mg/ml. Naïve tissues were used to prepare standard, quality control, and blank samples in tissue matrix. Homogenized tissue or plasma samples were kept on ice and spiked with internal standard. Analytes were extracted using protein precipitation techniques with acetonitrile and samples were then analyzed using a SCIEX Triple Quad 5500+ system – QTRAP ready linked to an ExionLC AD-HPLC system. Data analysis was performed using SCIEX OS software.

### Expression and purification of REV-ERBα LBD for X-ray crystallography

Gene of human *Rev-erbα* (P281-Q614) deleted from region P324-P424 was cloned into a peT-15b vector. The protein was produced in bacteria in fusion with a (His)$_6$ tag at the N-terminus with a thrombin cleavage site in between. The protein was overproduced in *Escherichia coli* BL21(DE3) strain in auto-induction medium at 25 °C during 2h30. Bacterial cells from 6 L were harvested by centrifugation and resuspended in lysis buffer (25 mM Na Hepes pH7.5, 150 mM NaCl, 2 M Urea). After 10-fold protein dilution in buffer A (25 mM Na Hepes pH7.2, 0.5 mM EDTA, 5 mM dithiotreitol (DTT), 5% glycerol) additional purification was performed with an ion exchange chromatography (HiTrap Q FF 5 ml) equilibrated in buffer A. A 30 CV-gradient from 0 to 1 M NaCl was applied, and protein eluted at a NaCl concentration of around 150 mM.

### Protein crystallization and structure refinement

Prior to crystallization, 2 molar equivalents of nuclear receptor corepressor 1(NCOR1) peptide (2045-THRLITLADHIAQIITQDFAR-2065) were added to REV-ERBα and the complex was concentrated to 5 mg/ml ultrafiltration. Crystallization experiments were carried out at 22 °C using the sitting drop vapor diffusion method in 96-well plates (300 nl protein solution + 300 nl reservoir) using Innovadyne nanodrop robot. Crystals grew in 80 mM Na Hepes pH7.5, 200 mM proline, 8% glycerol, and 18% PEG3350. Apo crystals appeared in 24 h. For soaking experiments, apo Reverbα/NCOR1 crystals were transferred to new drops of crystallization condition supplemented with $10^{-3}$M of STL1267 (M. wt 322.7 Da) for one month. After soaking, crystals were cryoprotected with 23% of glycerol added to the crystallization condition and flash frozen in liquid nitrogen. X-ray diffraction tests and data collections were performed on PROXIMA-1 beamline (Wavelength 0.978 Å) at SOLEIL synchrotron. X-ray data were processed with Autoproc/Staraniso. Molecular replacement and structure refinement were performed with Buster using PDB 3N00 as starting model. Ligand dictionary was generated with Grade. Final model quality was monitored by H (98.14% of residues in favored region).

### Molecular modeling

Molecular dynamics simulations were performed on the wild-type and F488A mutant of REV-ERBα/NCoR ID1/STL1267. Molecular dynamics simulations were performed on each system for one microsecond

(1 μs) using Amber18[49]. Analysis plots on the protein alpha carbons validating protein folding and the stability the simulations, such as root mean square deviation (RMSD), solvent accessible surface area (SASA), radius of gyration (Rg) is provided in Supplementary Fig. 6. Coordinates for the wild- type and mutant REV-ERBα/NCoR ID1/ STL1267 were obtained from the Xray structure. First, energy minimization was carried out using the steepest descent and conjugate gradient methods while keeping the ligand constrained. The constraints were removed and then each system was energy minimized entirely. The LeaP module was used to neutralize and solvate the complexes using an octahedral water box of TIP3P water molecules. The FF14SB forcefield parameters were used for all receptor residues and the general amber force field was applied to the ligand[50]. Nonbonded interactions were cut off at 10.0 Å, and long-range electrostatic interactions were computed using the particle mesh Ewald (PME). Ligands were modeled using Maestro and pictures were generated using UCSF Chimera and Maestro[51]. After energy minimization, the system was gradually heated with the Langevin thermostat to 300 K over 30 ps at constant volume using 1 fs time step. Initial velocities were sampled from the Boltzman distribution while keeping week restraints on the solute and the ligand. The system was then equilibrated in the isothermal-isobaric ensemble (NPT), at 300 K, using constant pressure periodic boundary with an average pressure of 1 atm. Isotropic position scaling was used to maintain the pressure with a relaxation time of 2 ps. The SHAKE algorithm was used to keep bonds involving H atoms at their equilibrium length. Two fs time step was used for the integration of Newton's equations. Production simulations were performed on GPUs using the CUDA version of Particle Mesh Ewald Molecular Dynamics (PMEMD) for one microsecond, 1 μs MD[52]. Distances were calculated between the center of mass of two selected residues. Distances, Rg, RMSD and SASA, RMSF calculations were calculated using CPPTRAJ[53]. The entropy and enthalpy calculations were performed using normal mode analysis approximation and MM-GBSA algorithm implemented in AMBER18 (Supplementary Table 4). Trajectory clustering as well as pictures and MD simulations movie were generated using UCSF Chimera[54]. Plots were generated using Gnuplot.

### Statistical analysis

Unless otherwise specified, statistical significance was determined by subjecting mean values per group to two-sided Student's $t$ test. A value of $p < 0.05$ is considered statistically significant.

### Reporting summary

Further information on research design is available in the Nature Portfolio Reporting Summary linked to this article.

## Data availability

The data that support this study are available from the corresponding authors upon reasonable request. Structural data have been deposited in the Protein Data Bank (PDB) under accession code 8D8I (REV-ERBα LBD and STL1267). Data underlying Fig. 2; and Supplementary Fig. 2 are available as a Source Data file. Source data are provided with this paper.

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

## Acknowledgements
This work was supported by a grant from the NIH (AG060769) to T.P.B. the Department of Defense office of the Congressionally Directed Medical Research Programs (CDMRP) W81XWH-19-1-0633 (to B.E.) and W81XWH-19-1-0632 (to T.P.B.). Funding was also provided by Pelagos Pharmaceuticals, Inc.

## Author contributions
T.P.B. and J.K.W. conceived the project. B.E., A.C, J.K.W., and G.B.V. synthesized STL1267. M.M. and A.C.V. performed FRET and reporter assays. M.M. performed gene expression studies and site directed mutagenesis. T.K. produced recombinant protein for the radioligand binding assay and performed this assay. F.C. and C.R. performed the X-ray crystallography study and analyzed the structure. L.H. performed the molecular simulations, conceptualization, analysis, discussion of the structure, molecular modeling and mutagenesis results. T.P.B., M.M., and L.H. wrote the manuscript with editing from all authors.

## Competing interests
T.P.B., J.K.W., and B.E. hold stock in Pelagos Pharmaceuticals, Inc. which provided partial funding for this research. Other authors have no competing interests.
