## [Peer Review File · Nature Communications]

Structural Basis of Synthetic Agonist Activation of the Nuclear Receptor REV-ERBReviewers' Comments:

Reviewer #1:

Remarks to the Author:

Overall, this is a commendable structure-function study on the transcriptional repressor, REV-ERBa. It includes the identification of an agonist that belongs to the non-heme class of small ligands. The study combines cell-based and in vitro, in vivo, and crystallographic analyses of the ligand-receptor molecular complex.

However, as a spring-board for therapeutic drug discovery, there are a few additional issues that may need to be clarified. These include,

1. In the HEK293 2-hybrid assay, the ERB agonists are described as activators of transcription. Perhaps, they would be more appropriately described as 'repressors'
2. Although the response of multiple NRs are tested for activity to STL1267, the constitutive androstane (CAR) and pregnane X receptors (PXR) have not. Since both CAR & PXR are strongly associated with xenobiotic metabolism, their role in STL1267 may be important, if this ligand is to be a viable therapeutic lead.
3. Associated with #3 above, is there any detection of changes in cytochrome P450 activity, or the activity of other xenobiotic metabolizing enzymes?
4. The F488a mutation increases NCoR binding. Does it affect STL1267 activity? The removal of the bulkier Phe sidechain may allow for the ligand binding pocket to wrap around the ligand more tightly, which may explain the higher activity of this mutant REV-ERB.
5. It is suggested that STL may alter the binding of corepressor, NCoR and associated pharmacology. Would this alteration extend to the recruitment of specific corepressor isoforms and transcriptional gene products?
6. A discussion of whether a 0.1-0.3micromolar potency in cell-based assays is sufficient or if there are specific approaches that can be taken to develop a higher level of potency towards drug-discovery.

Reviewer #2:

Remarks to the Author:

This manuscript authored by Murry et.al. is centered on characterizing small molecule modulators of REV-ERb, which plays a critical role in circadian rhythm, metabolism, inflammation, and other pathways. REV-ERb is a transcriptional repressor that has generated much excitement and current pharmacological tools to manipulate receptor activity are not ideal. The field would greatly benefit from potent, efficacious, and bioavailable chemical tools to aid in studying REV-ERb biology. This work is squarely focused on a recently reported agonist, STL1267, which shows higher affinity, efficacy, and improved biophysical properties relative to a published compound SR9009. The authors compare efficacy, potency, basic NHR mechanism, and off-target pharmacology for NHRs and receptors/channels important for the central nervous system. They also report a crystal structure of the REV-ERb-STL1267-NCoR-ID1 peptide complex paired with molecular dynamics and mutagenesis analysis to provide some insight into how SRL1267 drives agonism focusing largely on which (aromatic) residues within the ligand binding pocket are involved in binding and receptor activity. This is the first crystal structure of REV-ERb with a synthetic (non -porphyrin or heme agonist).

This is an important topic, but the manuscript seems rushed with numerous grammatical errors and lacks sufficient detail and description for several figures and associated legends. Typos are numerous; however, the meaning of sentences are not obscured.

The PDB validation report is missing (along with key details) which does not allow for a proper review of the quality of the X-ray structure and subsequent interpretations.

Does 1.8 vs 4.7 micromolar represent a 'much' greater potency for STL1267 vs SR9009.

Figure 1. the number of technical replicates for the FRET assays are not given. Biological replicates are not stated for the cell-based assays.

Extended Data 2: Were male and/ or female mice used?

There is no data or methods presented for the MS (or LC/MS/MS??) based detection of STL1267.

SPA assay: How was a K_i calculated without first determining a K_d for the radiolabeled STL1267? This is required. The text states that tritiated STL1267 was used in excess (saturated beads).

SPA assay: How is it possible that the optical properties of the HEME interfere with the radioligand binding assay?

It seems than an unbiased gene expression analysis (e.g. RNAseq) would better compare the activity and gene expression activity of these ligands.

Figure 3. Electron density is shown for the ligand but is not described in the legend. What is the contour level? What type of map is shown (omit, composite omit, 2Fo-Fc)? Also, a close-up view of the ligand in the electron density should be shown.

For all structural figures, the N- and C-termini should be indicated along with breaks in the model (such as between H1 and H3). All colors should be explained.

Figure 4. Is the sphere representing the Iron?

Extended Data 4: A legend is needed to explain the figure! What is blue and what is green (bound ligand)?

Extended Data 4: Most concerning is that the alpha carbon of M520 appears to have 6 bonds. This is a major error and would affect the simulation.

Extended Data 4: Overall, the figure is challenging to interpret, what is the author trying to highlight? Perhaps removing the hydrogens would help.

Molecular Dynamics: Were any replicate simulations run? 1 microsecond is certainly sufficient, and I am not asking for replicate 1 microsecond runs as long as the simulation was stable.

Table 1: Unit Cell Dimensions should have 1 sig fig.

We thank the reviewers for the constructive comments, and we have responded to each of the comments in red below.

Reviewer #1 (Remarks to the Author):

Overall, this is a commendable structure-function study on the transcriptional repressor, REV-ERB α . It includes the identification of an agonist that belongs to the non-heme class of small ligands. The study combines cell-based and in vitro, in vivo, and crystallographic analyses of the ligand-receptor molecular complex.

However, as a spring-board for therapeutic drug discovery, there are a few additional issues that may need to be clarified.

These include,

1. In the HEK293 2-hybrid assay, the ERB agonists are described as activators of transcription. Perhaps, they would be more appropriately described as 'repressors'.

We have updated the manuscript to clarify this issue. In the original manuscript, we used the following to describe how the 2-hybrid assay switches agonists from repressors to activators to allow for better resolution of activity, "This format of the assay essentially functions as a mammalian 2-hybrid assay and detects REVERB agonists as activators of transcription effectively enhancing the sensitivity of such an assay from the typical one hybrid system that detects enhancement of transcriptional repression."

We have modified this to be clearly identify agonists as repressors of transcription, "This format of the essentially functions as a mammalian 2-hybrid assay and detects REVERB agonists (that function as repressors of transcription) as activators of transcription, in the context of this assay, effectively enhancing the sensitivity of such an assay from the typical one hybrid system that detects enhancement of transcriptional repression."

2. Although the response of multiple NRs are tested for activity to STL1267, the constitutive androstane (CAR) and pregnane X receptors (PXR) have not. Since both CAR & PXR are strongly associated with xenobiotic metabolism, their role in STL1267 may be important, if this ligand is to be a viable therapeutic lead.

As suggested by the reviewer, we have performed the CAR and PXR assays. STL1267 did not have activity at CAR but had some detectable activity at PXR. STL1267 displayed 18% of the activity of 5 μ M rifampicin. This is now included in the specificity table.

3. Associated with #3 above, is there any detection of changes in cytochrome P450 activity, or the activity of other xenobiotic metabolizing enzymes?

We have run STL1267 in a mouse liver microsome stability assay and it had a half-life of only 34 minutes indicating it is rapidly metabolized. Based on these data, this particular compound would not progress towards further evaluation, so we did not conduct detailed metabolism assessments. We have added this information to the manuscript.

4. The F488a mutation increases NCoR binding. Does it affect STL1267 activity? The removal of the bulkier Phe sidechain may allow for the ligand binding pocket to wrap around the ligand more tightly, which may explain the higher activity of this mutant REV-ERB.

The F488A mutation does not affect basal NCoR binding to REV-ERB much, rather it only affects STL1267-dependent recruitment of NCoR which we use as a surrogate of STL1267 activity. We have not probed the effect on any other markers of REV-ERB activity. We agree with your mechanistic view of the F488A mutation and have added additional details to the manuscript.

5. It is suggested that STL may alter the binding of corepressor, NCoR and associated pharmacology. Would this alteration extend to the recruitment of specific corepressor isoforms and transcriptional gene products?

This is entirely possible, and we have added some detail to the discussion to reflect this possibility.

6. A discussion of whether a 0.1-0.3µM potency in cell-based assays is sufficient or if there are specific approaches that can be taken to develop a higher level of potency towards drug-discovery.

As suggested by the reviewer, we have added additional discussion concerning the range of potencies that may be sufficient for drug development for REV-ERB agonists as well as general methods for improving both PK and PD parameters.

Reviewer #2 (Remarks to the Author):

This manuscript authored by Murry et.al. is centered on characterizing small molecule modulators of REV-ERb, which plays a critical role in circadian rhythm, metabolism, inflammation, and other pathways. REV-ERb is a transcriptional repressor that has generated much excitement and current pharmacological tools to manipulate receptor activity are not ideal. The field would greatly benefit from potent, efficacious, and bioavailable chemical tools to aid in studying REV-ERb biology. This work is squarely focused on a recently reported agonist, STL1267, which shows higher affinity, efficacy, and improved biophysical properties relative to a published compound SR9009. The authors compare efficacy, potency, basic NHR mechanism, and off-target pharmacology for NHRs and receptors/ channels important for the central nervous system. They also report a crystal structure of the REV-ERb-STL1267-NCoR-ID1 peptide complex paired with molecular dynamics and mutagenesis analysis to provide some insight into how SRL1267 drives agonism focusing largely on which (aromatic) residues within the ligand binding pocket are involved in binding and receptor activity. This is the first crystal structure of REV-ERb with a synthetic (non -porphyrin or heme agonist).

This is an important topic, but the manuscript seems rushed with numerous grammatical errors and lacks sufficient detail and description for several figures and associated legends. Typos are numerous; however, the meaning of sentences are not obscured.

We have edited the manuscript to correct grammatical errors.

The PDB validation report is missing (along with key details) which does not allow for a proper review of the quality of the X-ray structure and subsequent interpretations.

Validation of the PDB report is enclosed

Does 1.8 vs 4.7 micromolar represent a 'much' greater potency for STL1267 vs SR9009.

We used the term "much greater potency" in the discussion without referring to the assay context. The values indicated above refer to the cell-based 2-hybrid assay. In the radioligand binding assay, 1267 is 4.3-fold more potent than 9009. In the FRET assay, 1267 was 14.6-fold more potent than 9009. Based on this comment, we decided to remove "much".

Figure 1. the number of technical replicates for the FRET assays are not given. Biological replicates are not stated for the cell-based assays.

Both FRET and cell-based assays are performed in triplicate and typically repeated 3 times. This information is has been added to the figure legend.

Extended Data 2: Were male and/ or female mice used?

There is no data or methods presented for the MS (or LC/MS/MS??) based detection of STL1267.

Our methods section indicates that male mice were used, "C57Bl/6J mice (male; 6-8 weeks of age) were injected IP with 50 mg/kg STL1267 (n=18) or vehicle (10% DMSO, 12% TWEEN80, PBS) (n=12)."

We have added the methods for the LC/MS/MS detection for STL1267 to the methods section as requested by the reviewer.

SPA assay: How was a K_i calculated without first determining a K_d for the radiolabeled STL1267? This is required. The text states that tritiated STL1267 was used in excess (saturated beads).

We did indeed calculate the K_d for ^3H -STL1267 binding for REV-ERB α and that value is 448 nM. We have added this information to the manuscript. ^3H -STL1267 was used in the competition experiments at ~25% of the level of saturation. The manuscript has been updated to provide these details.

SPA assay: How is it possible that the optical properties of the HEME interfere with the radioligand binding assay?

Radioligand binding assays use scintillant to detect the β particle decay, thus there is a critical optical component the assay. If the β particles released by tracer are close enough to the SPA beads the scintillant within the beads releases light which is then detected in a scintillation counter. The scintillant emits light within a broad range of wavelengths 350-500 nm. This overlaps considerably with the strong optical properties of heme and renders the assay impractical when heme is present.

It seems that an unbiased gene expression analysis (e.g. RNAseq) would better compare the activity and gene expression activity of these ligands.

We agree that a “global” analysis of gene expression in response to 1267 vs. 9009 would be useful to identify unique profiles of these ligands. However, our intention was to compare STL1267 to some well characterized SR9009 regulated genes in this first report of the novel REV-ERB agonist chemotype and X-ray structure. We are in the process of performing a large global gene expression comparison between 1267 and 9009 in multiple cell lines in WT and REV-ERB KO cell lines and this will be the subject of a future manuscript.

Figure 3. Electron density is shown for the ligand but is not described in the legend. What is the contour level? What type of map is shown (omit, composite omit, 2Fo-Fc)? Also, a close-up view of the ligand in the electron density should be shown.

As suggested by the reviewer, a separate figure for the ligand electron density was added to figure 3d and related information is available in the figure legend.

For all structural figures, the N- and C-termini should be indicated along with breaks in the model (such as between H1 and H3). All colors should be explained.

All figures' legends were revised and all colors were explained. N- and C-termini between H1 and H3 were included in all relevant figures.

Figure 4. Is the sphere representing the Iron?

Yes, the orange sphere represents the iron. We have added additional information to the figure legend.

Extended Data 4: A legend is needed to explain the figure! What is blue and what is green (bound ligand)?

This figure illustrates the molecular interactions of Phe488 after the MD simulations. A legend was added. Green corresponds to the protein and bound ligand before the simulations, blue corresponds to the protein and bound ligand after the simulations. The protein and bound ligand before the simulations were deleted from the picture for simplicity.

Extended Data 4: Most concerning is that the alpha carbon of M520 appears to have 6 bonds. This is a major error and would affect the simulation.

The alpha carbon of M520 does not have four hydrogens, it has two hydrogens, behind the alpha carbon is the beta carbon with another two hydrogens, due to the angle that the picture was taken at it, made it seem as if the alpha carbon had 4 hydrogens.

Extended Data 4: Overall, the figure is challenging to interpret, what is the author trying to highlight? Perhaps removing the hydrogens would help.

This picture is a closer view of the molecular interactions of Phe488. This residue changed its conformation and flipped outside of the LBP during the simulations (Figure 6B) and formed

hydrophobic interactions with side chains of amino acid residues Phe521, Leu491 and Leu505 (Extended Data Fig. 4). This information is discussed in the text. We removed all non-polar hydrogens.

Molecular Dynamics: Were any replicate simulations run? 1 microsecond is certainly sufficient, and I am not asking for replicate 1 microsecond runs as long as the simulation was stable.

We did not run replicate 1 μ s simulations. We ran 10 independent 100 nanosecond simulations for a total of 1 microsecond simulation. Yes, the simulations were stable. The RMSD for the protein (non-hydrogen atoms) is less than 3Å all over the simulations.

Table 1: Unit Cell Dimensions should have 1 sig fig.

Changed as the reviewer suggested.

Reviewers' Comments:

Reviewer #1:

Remarks to the Author:

The authors have made a commendable effort to address all the concerns of the review of their original submission. At this stage, I have no more criticism to offer.

Reviewer #2:

Remarks to the Author:

I am satisfied with the authors critiques.

Reviewer #3:

Remarks to the Author:

The study is focus on how ligands bind to and 53 regulate transcriptional repression by REV-ERB based on the structure of heme bound to REV54 ERB.

The authors characterizes a high affinity synthetic REV-ERB agonist, STL1267, and described its mechanism of 58 binding to REV-ERB as well as the the experimental and theoretical methods. The work is innovative. however the it is need to be major revised to be published in NC.

Q1 Which forcefield did the authors used for the heme (contained Fe 2+) ?

In Paragraphs 601-602 the authors pointed out the FF14SB forcefield parameters were used for all receptor residues and the general amber force field was applied to the ligand51 602 . Did FF14SB forcefield used for heme?

Q2 RMSD Rg SASA should be submitted as Supplementary document.

Q3 What is the conformational changes for the ligand binding to the WT and mutant?

Q4 the free energy of binding should be added (MM-PBSA, Ti or FEP)

We thank the referees for their comprehensive review of the manuscript and are pleased that we have satisfied the original 2 referees concerns. In this revision, we have addressed the concerns of the 3rd referee that was added in after the 1st revision was received. Our responses to the referees' comments are in red below:

Reviewer #1 (Remarks to the Author):

The authors have made a commendable effort to address all the concerns of the review of their original submission. At this stage, I have no more criticism to offer.

Reviewer #2 (Remarks to the Author):

I am satisfied with the authors critiques.

We thank both reviewers for their comments and aid in improving our manuscript.

Reviewer #3 (Remarks to the Author):

The study is focus on how ligands bind to and regulate transcriptional repression by REV-ERB based on the structure of heme bound to REV-ERB.

The authors characterizes a high affinity synthetic REV-ERB agonist, STL1267, and described its mechanism of binding to REV-ERB as well as the experimental and theoretical methods. The work is innovative. however, the it is need to be major revised to be published in NC.

We would like to reinforce that the focus of this manuscript is to describe a novel synthetic, non-porphyrin, agonist of REV-ERB and describe the mechanism of binding based on the X-ray crystal structure of this synthetic ligand to the REV-ERB ligand binding domain. This is the first structure of a non-porphyrin REV-ERB ligand. Our focus was not to use a heme-REV-ERB structure to define how ligands regulate transcriptional repression.

Q1 Which forcefield did the authors used for the heme (contained Fe 2+) ?

In Paragraphs 601-602 the authors pointed out the FF14SB forcefield parameters were used for all receptor residues and the general amber force field was applied to the ligand51 602. Did FF14SB forcefield used for heme?

In the current manuscript, we report the Xray structure of a high affinity synthetic REV-ERB agonist, STL1267, and compare its mechanism of binding to REV-ERB to the Xray structure of heme-bound REV-ERB (PDB: 6WMQ).

Molecular dynamics simulations were not performed on the REV-ERB bound heme. Molecular dynamics (MD) simulations were performed only on our newly reported REV-ERB bound STL1267 only for the purposes of investigation of why the F488A mutation led to a gain of function. We have added information in the methods and results sections to make this clear for readers and avoid confusion.

Q2 RMSD Rg SASA should be submitted as Supplementary document.

As requested by the reviewer, these plots are now included in the supplementary document (Extended Data Fig. 6).

Q3 What is the conformational changes for the ligand binding to the WT and mutant?

In response to this question, we performed an additional one microsecond of molecular dynamics simulations of the F488A REV-ERB mutant bound with STL1267 to investigate their conformational changes and to calculate ligand binding free energies. The ligand binding pose was stable in both the wild type and mutant during the simulations. In the case of the wild type protein, the F488 side chain altered its conformation and its side chain flipped outside of the pocket (Figure 4). This is in accordance with the mutagenesis experiment where mutation of this amino acid residue did not abolish ligand binding. In the case of the MD simulations for the F488A mutant, no conformational changes were observed (Extended Data Fig. 5).

Q4 the free energy of binding should be added (MM-PBSA, Ti or FEP)

As requested by the reviewer, we have added this information. We carried out relative binding free energy calculations (Extended Table 4). In the case of the F488A mutant, both the enthalpy (ΔH) and entropy (ΔS) contribution of STL1267 to the total binding free energy were more favorable (ΔG value is -24.4 Kcal/mol for the mutant versus a ΔG value of -20.8 Kcal/mol for the wild type). These results are in accordance with the mutagenesis experiments where F488A mutation led to a gain of function.

Reviewers' Comments:

Reviewer #3:

I think this study can be accepted at this version.